# Research and validation of forest carbon sequestration measurement model based on biomass method-A case study of Guizhou Province

**Zhang Min[1,2], Wu Yang[1,2]\*, Qiu Yan[1,2], Yan Jun[1,2], Li Chunling[1,2], Ran Wen Rui[3]**

**1** School of Resources and Environmental Engineering, Guizhou Institute of Technology, Guiyang, China, **2** Engineering Research Center of Carbon Neutrality in Karst Areas, Ministry of Education, Guiyang, China, **3** Guizhou Natural Resources Survey and Planning Research Institute. Guiyang, China

\* yangwu@git.edu.cn

## Abstract

With the establishment of China Certified Voluntary Emission Reduction (CCER) market, more and more people pay attention to the development of forestry carbon sequestration in Guizhou. Guizhou is rich in forestry resources, but at the current stage, there are a series of measurement difficulties in the development of forestry carbon sequestration. Some traditional research methods have the problems of heavy workload and over-reliance on manual work. In view of the above problems, this paper studies the construction of carbon sequestration measurement model based on biomass estimation model. In this paper, sampling design, project boundary, carbon pool selection and other factors are considered in the construction of the model. At the same time, in view of the problem that the calculation results of the actual net carbon sequestration caused by $CH_4$, $N_2O$ and other gas emissions in forestry carbon sequestration projects are not accurate enough, this study deduces a specific calculation model based on the actual project development needs and the data of 10 forest farms that have formed forestry carbon tickets. Taking Chinese fir as the research object, the effect of carbon sequestration model was verified and analyzed. Compared with the existing forestry carbon sequestration project evaluation, the results showed that the average relative error of the model was 6. 09%, and the absolute error range was 0. 348-4. 262/hm², and the model effect was good. The establishment of the model not only solves the problems of over-reliance on manpower, material resources and inefficiency in the development of forestry carbon sequestration in China, but also accelerates the rapid and vigorous growth of forestry carbon sequestration industry in Guizhou, and contributes China's wisdom and effective solutions to global climate governance.

## Introduction

With a series of natural disasters and economic losses caused by climate warming, the international community calls on all countries in the world to jointly manage the environment in

**Data availability statement:** All relevant data are within the manuscript and its Supporting Information files.

**Funding:** This research was supported by the Concealed Ore Deposit Exploration and Innovation Team of Guizhou Colleges and Universities (Guizhou Education and Cooperation Talent Team [2015] 56), Provincial Key Discipline of Geological Resources and Geological Engineering of Guizhou Province (ZDXK [2018] 001), Huang Danian Resources of the National Colleges and Universities Teachers' Team of Exploration Engineering (Teacher Letter [2018] No. 1), Geological Resources and Geological Engineering Talent Base of Guizhou Province (RCJD2018-3), Key Laboratory of Karst Engineering Geology and Hidden Mineral Resources of Guizhou Province (Qianjiaohe KY [2018] No. 486 Guizhou Institute of Technology Rural Revitalization Soft Science Project [2022xczx10]), and the Education and Teaching Reform Research Project of Guizhou Institute of Technology (grant numbers JGZD202107 and 2022TDFJG01). the sponsors or funders do not play any role in the study design, data collection and analysis, decision to publish, or preparation of the manuscript There was no additional external funding received for this study.

**Competing interests:** NO authors have competing interests.

accordance with the conventions and systems of the Climate Conference. In the process of studying this issue, the research on forestry carbon sinks has also attracted much attention [1]–[5]. Driven by a series of climate summits, investment in carbon sink construction, mainly for developing countries, is increasing. According to the Global Forest Resources Assessment Report, China is the country with the largest number of plantations and the fastest growth of forest area in the world from 2010 to 2020. With such huge forest resources, the development of forestry carbon sequestration in China also has great potential. According to the official website of the Guizhou Provincial Forestry Bureau.As a pioneer in the construction of ecological documents, Guizhou is rich in forestry resources, with a forest area of 169 million mu, a forest area of 166 million mu and a forest coverage rate of 62.81%, ranking fifth in the country. There are 105 state-owned forest farms with a total management area of 5.42 million mu, 5 national forest cities, 48 provincial forest cities, 413 forest townships, 1963 forest villages, 9058 forest households and 52 beautiful forest villages. The green coverage rate of villages in the province reached 46.24%. At present, the measurement problems encountered in the development of forestry carbon sequestration are still in the research stage. Some traditional research methods have the problems of heavy workload and over-reliance on manual work, such as the representative reserve change method and profit and loss method [6]. The change in reserves method evaluates the annual change in carbon stocks over time [7]. This method obtains basic data, mainly forest volume, through a large number of ground surveys (such as the National Forest Inventory) [8]. Then the carbon storage of each carbon pool of the ecosystem is estimated by using the established biomass model and carbon measurement parameters (forestry industry standards or related literature), or related equations, biomass carbon density and soil organic carbon density, etc. Finally, the annual change of carbon storage in a period of time is analyzed. If the annual variation of carbon storage is positive, it is regarded as a "carbon sink," otherwise, it is regarded as a "carbon source." Up to 2023, China has carried out several continuous forest resource inventories at intervals of 5 years, and many provinces and regions have also carried out the second class forest resource inventory for 10 years. These survey data and statistical data are the basis of this method.The profit and loss method is to assess the carbon sink function of the ecosystem by studying the carbon budget of the ecosystem in a certain period of time and using the balance results of carbon absorption and emission. This requires a lot of manpower to measure the biomass in the field and establish a regression model with DBH, tree height, volume, forest age and other variables to describe the growth rate. This method can also be used to assess and predict the impact of climate and other environmental factors on growth and carbon accumulation. Commonly used models include FVS [9,10], CACTOS [11], ORGANON [12] and TreeGross [13]. Full carbon pool models mainly include CO2FIX [14], TreeGroSS-C [15], FORCARB [16] and CBM-CFS3 [17]–[18].The reserve change method requires that the two survey data are obtained by using the same method, using the same biomass model and parameters, and the data based on continuous sample plots are the most accurate and reliable, but the workload is large, and the accuracy can only be guaranteed on the scale of the sampling population. The profit and loss method obtains the forest carbon sink through the empirical statistical model and the process model, but it has strong regional characteristics, so it needs to re-determine the empirical parameter values or the localized process model parameters when it is applied in other regions, and the parameters of the process model are usually not easy to obtain, which limits the use of the model. On the other hand, there is no clear specification for the measurement methods of forest carbon sinks at present. The existing measurement methods mainly include biomass method, volume method and eddy correlation method. Among these methods, the biomass method has high practicability because of its simple and convenient operation, so this paper mainly studies the application of biomass method in the theoretical study of forestry carbon sequestration.The premise of using biomass

method as forest carbon sink measurement method is to estimate forest biomass more accurately and quickly. Among many technical schemes, the forest land survey method is widely recognized by forestry personnel in China, and its measurement tools and methods have been mature in China. The traditional binary regression model is the most frequently used method in the current forest land survey method. The calculation model of this method is simple, but the model needs to assume in advance and select the geographical area. The model of the same tree species in different geographical areas is different, and the accuracy of biomass estimation is relatively low [19]. Therefore, this paper attempts to use the BP neural network prediction method to estimate the forest biomass quickly. The model not only has a certain generalization ability, but also takes into account the data error of forest growth caused by different geographical locations, which can expand new ideas, methods and technical support for the estimation of forest biomass in Guizhou.

## Data collection and preprocessing

### Data collection

The data in this paper mainly come from the forest resources inventory, Guizhou Forestry Bureau, Guizhou Statistical Bureau and related forestry literature. The data include area, number of trees, volume, stand type, canopy density, ecological location, tree species composition, forest slope aspect, forest slope, average forest age, shrub coverage, number of trees per hectare, initial planting density per hectare, scattered tree species, growth and other survey factors. Human activity had little impact in the selected study area and growth was stable over time. Exclude special circumstances such as forestry activities, fruit tree planting and being in the urban green belt in the study area. Finally, the collected data is divided into two parts, which are used for training and simulation.

### Data preprocessing

Every forest resource inventory in China needs tens of thousands of staff, because it is manual operation, there are often problems such as improper use of equipment, and improper use of equipment will lead to abnormal data, so in order to ensure the accuracy of data analysis and use in the later period and improve the accuracy of modeling, it is necessary to preprocess the collected data.

### Data harmonization

The purpose of data unification is to unify the format and name fields of the data to be analyzed, such as the unification of different measurement units and species information. For the unification of measurement units, the professional forestry involved in this study, such as the three parameters of biomass, diameter at breast height and tree height, are measured in kg, cm and m respectively, while the geographical location is replaced by the number 0–5, representing six regions. If the forestry industry indicators involved are not clearly defined, the national standards shall be uniformly used. The species information is unified, and the names of trees in different regions are inconsistent. For example, the Chinese fir studied in this paper has many names, such as sand wood, sand tree, thorn fir, cedar, etc. In view of the above situation, the names of tree species participating in the model training are unified by consulting the scientific names of tree species recorded in the Flora of China

### Data splicing

Data splicing refers to merging the statistical data of multiple Excel tables and merging the fields according to the same term. All the Excel table data are spliced and merged. This step

is a crucial step to ensure the accuracy of subsequent data analysis. In the process of data processing, it is impossible to determine whether the normal value is an abnormal factor. By analyzing the relationship between the factor and other factors, such as the linear or nonlinear relationship between diameter at breast height and biomass, geographical location and biomass, the abnormal value is derived by using the relevant model constructed by historical documents. If the derived abnormal value is within the error range, it will be retained. If the error range is exceeded, the abnormal value can be deleted directly.

### Data cleaning

The main purpose of data cleaning is to clean up some abnormal data caused by different documents or standards. The accuracy of the biomass data field determines the accuracy of the model in this paper, so all the data of a single tree can be deleted directly if the biomass data is missing. For the specific missing data of different organs of trees, the calculation can be completed according to the existing tree measurement methods and relevant national standard calculation formulas. For the research data, some data are measured by forestry monitoring personnel on the spot, and there may be errors in the process of recording, which may lead to a large deviation between the data and the actual value, so these outliers need to be discarded.

## Research and verification of carbon sequestration measurement model based on biomass method

Biomass method, relaxation eddy accumulation method, eddy correlation method and so on are all the measurement methods of forestry carbon sink, among which the biomass method has high practicability because of its convenient operation, but the current biomass method measurement model is complex and redundant because of the large number of forest miscellaneous trees and complex calculation steps. Therefore, this question aims at the forestry carbon sequestration project, and deduces the forestry carbon sequestration measurement model of biomass method [19–20].

### Project boundaries

**Project boundary concept.** Geographic area and time horizon are two major considerations for project boundaries in carbon sink development. Therefore, the project boundary can be divided into geographical boundary and time boundary.

Geographical boundary: In the forestry carbon sequestration project, the geographical area is not limited to one, and multiple areas are allowed to develop the project at the same time. The geographical boundary of the project needs to be determined statistically by using the corresponding method.

Time boundary: The project approved by the National Development and Reform Commission can enter the project implementation stage, and the project will be timed from the beginning to the end of the project.

**Determination method of project boundary.** Determination of geographical boundaries; In the process of project development, it is necessary to clearly define the geographical location of the project area to facilitate the later verification of the net carbon sink generated by the project. With the development of science and technology, there are many methods to determine the geographical boundary of the project. In summary, one of the following methods can be selected to determine the boundary:

1) Using the Beidou Navigation Satellite System (BDS) or other satellite positioning systems such as GPS, the geographical inflection point boundary can be directly measured in the field of the project plot.

2) Use the geographic topographic map to go to the forest land for field delineation, and then use the Beidou Navigation Satellite System (BDS) for further accuracy control.

3) Forestry regionalization at county or township level.

Determination of time boundary: Forestry carbon sequestration projects usually use the start and end time to determine the time boundary, and the time boundary is generated according to the time boundary. At present, the domestic carbon market has been opened, emission control enterprises have been included in the scope of supervision, and more and more enterprises have joined the carbon market, which, to a certain extent, promotes project developers to flexibly use crops that have become useful in a short time in forestry carbon sequestration projects. If the carbon market is depressed, tree species with high crop value and slow growth can be selected.

## Determination of carbon source and carbon pool index

The IPCC has made a specific description of the specific carbon pools involved in forestry carbon sinks, as shown in Table 1.

## Carbon pool

In the process of forestry carbon sequestration project development, if we want to obtain the exact amount of carbon sequestration generated by the project, we need to select and measure all the carbon pools, but this leads to a great increase in the labor cost of measurement and accounting in the process of project development. Therefore, in the actual development process, we can choose the corresponding carbon pool conservatively. If the increase of some carbon pools in the forestry carbon sink development project is very small, we can choose not to measure and account the carbon pool with small increase. In this way, certain metering costs can be reduced. The proportion of carbon sink in different terrestrial ecosystems is different, for example, the biomass on artificial forest land is more than that on farmland, and the carbon storage of soil carbon pool in grassland land use system is huge, so it is necessary to consider the land use system comprehensively in the process of carbon pool selection. The carbon pool of litter or dead wood did not exist in the grassland system. For all forestry carbon sinks, the aboveground biomass carbon pool is the most important and must be selected in the process of measurement.

The carbon pool is selected according to the comprehensive consideration of the actual development. The less the carbon pool is selected, the lower the measurement cost is, and the greater the benefit of the final project is, which will attract more owners to invest in the development of carbon sinks. This will not only improve the enthusiasm of project development and generate economic value, but also help the healthy and rapid development of China's low-carbon economy. Therefore, in the actual development process, the measurement cost of the whole project should be far less than the carbon sink value generated by the project

Table 1. Classification and description of carbon pools.

| Biomass | Aboveground biomass | Every living plant on the ground. |
|---|---|---|
| | Underground biomass | Total biomass of all living roots, with defined range |
| Dead organic matter | Dead wood | Dead roots and trees with diameter $\geq$ 10 cm |
| | Litter | Litter that naturally falls from dead parts of plants into the soil. |
| Soil | Soil organic carbon | Carbon storage in the soil of the whole woodland |
| Products | Agricultural products | Products formed during the growth of crops |

to ensure the smooth development of the project, so some carbon pools can be ignored in the development. In the project that does not consider the cost, because the increase of carbon pool causes the increase of project cost, the carbon sink measurement of the actual project is not accurate enough, which is not in line with the project requirements. By combining with the actual development situation, there are some projects that do not need to ignore the carbon pool in the actual project, so we can choose the carbon pool according to the actual project requirements. The proportion of carbon sink in the actual project will also change according to the growth of trees. Zhou Yurong [22] calculated the biomass carbon storage of forest ecosystem in China in 1993, and the results showed that the litter carbon storage accounted for only 3.1% of the total forest carbon storage, while the soil carbon storage accounted for 74.8% of the total forest carbon storage. Cheng Tangren et al. [23] studied the proportion of forest biomass and concluded that the carbon storage of litter carbon pool was negligible. Huang Congde [24] calculated the carbon storage of the plantation ecosystem, and estimated that the soil layer was the largest and the shrub and grass layer was the smallest. Accounting for 11.14%, 0.32%, 0.66% and 87.88% of the total carbon, respectively. Wei Wenjun [25] studied the forest ecosystem of Dagang Mountain in Jiangxi Province and concluded that the carbon densities of tree layer, understory vegetation, litter and soil were $3.136 \text{kg/m2}$, $0.529 \text{kg/m2}$, $0.324 \text{kg/m2}$ and $11.564 \text{kg/m2}$, respectively. Hong Tao et al. [26] studied the change law of carbon storage and carbon pool allocation of Aleurites Montana plantation in Fujian Province, and the results showed that the soil carbon storage of each stand increased with the increase of tree age.

Through the study, it is found that for forestry carbon sequestration projects, the proportion of dead leaves and branches is very small, and they are generally not measured. The carbon storage in the soil carbon pool is huge, but the soil carbon storage has a strong stability in the whole process of forestry carbon sink development, and the carbon storage will not fluctuate too much. Zhang Xiaoquan [27] studied the change of carbon storage in soil in a certain period, and the results showed that the change of biomass carbon storage was mostly less than 2%, and the fluctuation was very small.

Therefore, considering the cost effectiveness of the participants in the actual development process of the project, only the aboveground and underground biomass carbon pools can be selected to measure the carbon sequestration amount of the forestry carbon sequestration project.

## Selection of GHG emission sources

In forestry carbon sink development projects, sudden forest fires will cause tree burning, resulting in $CO_2$, $CH_4$ and $N_2O$ greenhouse gas emissions. If there is no fire during the development of the project, this part can be ignored. If biomass combustion occurs, it will cause $CO_2$, $CH_4$ and $N_2O$ emissions, so the three gas emission sources can not be excluded in case of fire.

## Sampling design

**Sample plot quantity calculation.** Geographic For forestry carbon sink development projects, the statistical method of sampling can be used to estimate the change of carbon storage in the pre-selected carbon pool. In the actual forestry carbon sequestration development projects, there are also certain requirements for measurement accuracy, which needs to reach 90% accuracy under 90% reliability level. In the development of forestry carbon sequestration projects, there may be too few sample plots in the actual project. If there are too few sample plots, more sample plots can be extracted within the project boundary to meet the

requirements of accurately measuring carbon sequestration. Through research, the number of sample plots for project development can be derived by the following methods.

If the number of sample plots calculated according to Formula 1 is greater than or equal to 30, the calculation result is more appropriate and can be used as a parameter for the actual project; If the calculated number of sample plots is less than or equal to 30, the second iteration calculation shall be carried out by using Formula (1) [21].

$$n = \frac{N * t^2_{VAL} * \left(\sum_i W_i * S_i\right)^2}{N * E^2 + t^2_{VAL} * \sum_i W_i * S_i^2} \tag{1}$$

Where: n is the number of monitoring plots, N = A/Ap, A is the total area of the project, and Ap is the area of the plots.

$t_{VAL}$ is the reliability index, $W_i$ is the area weight, and $S_i$ is the standard deviation of biomass carbon stock estimates

E is the absolute error range of the carbon stock estimate, 1, 2, 3 It's the carbon layer of the project

Calculate the number of sample plots according to Formula (1). If the area of sample plots is less than 5%, the n value calculated by formula (1) shall be adjusted with formula (2) to obtain the final number of sample plots (n a).

$$n_a = n * \frac{1}{1 + n / N} \tag{2}$$

Among $n_a$ For The number of monitoring plots after adjustment, and N is the sampling population of the testing plots.

When the target area is small, the simplified formula (3) can be used to calculate the number of sample plots, but the formula 1.3 is applicable to the situation that the area of sample plots is less than 5%.

$$n = \left(\frac{t_{VAL}}{E}\right)^2 * \left(\sum_i W_i * S_i\right)^2 \tag{3}$$

Where n is the number of monitoring plots, and $t_{VAL}$ is the reliability index.

In the process of calculating the number of detection plots in each layer, the optimal allocation method can be used to calculate the results, as shown in formula (4).

$$ni = n * \frac{W_i * S_i}{\sum_i W_i * S_i} \tag{4}$$

Where $n_i$ is the estimated number of monitoring plots.

**Sample plot setting.** During the implementation of the project, the final measurement results of the carbon storage of the carbon pool are calculated by setting up sample plots. It is necessary to ensure that the forest management methods inside and outside the sample plot are completely consistent. Where the selected carbon horizon contains multiple project monitoring plots, the plots should be distributed as evenly as possible.

The boundary line of the circular sample plot is prone to be overmeasured or missed in the actual surveying and mapping, which is not conducive to the precise control of the carbon sequestration of the project. On the other hand, the forest land in China is mostly located in complex terrain, so the circular sample plot is not suitable. Relatively speaking, the rectangular or square sample plot is more suitable for the carbon sequestration project in China.

**Layers of project zones.** Land within the project boundary may have an uneven distribution of biomass due to differences in soil conditions and use patterns. This situation results in increased costs, so a tiered approach can be used to reduce monitoring costs.

The specific arrangement of project stratification shall be comprehensively considered from the following points:

Original state of the land: the state of the land at the beginning of the project;

Tree species: distinguish according to different afforestation tree species;

Management mode: density of afforestation, etc.

Activities carried out within the project boundary: regeneration of natural forests or afforestation of trees in rotation, etc.

## Carbon sink measurement model

Carbon sink measurement is the estimation of the change of carbon storage in a selected carbon pool within the time boundary. Sampling design, project boundary, carbon pool selection and other factors need to be considered when building the model. At the same time, in view of the problem that the calculation results of the actual net carbon sequestration caused by $CH_4$, $N_2O$ and other gas emissions in forestry carbon sequestration projects are not accurate enough, a specific calculation model formula may be given to ensure the accuracy of the results. The net carbon sink generated by the project is calculated as shown in formula (5):

$$C_{P,t} = \Delta C_{P,i} - GHG_{E,t} - LK_t - \Delta C_{BSL,t} \tag{5}$$

Among $C_{P,t}$ it is the net carbon sink of the project. $\Delta C_{P,t}$ is the change in the carbon stock of the project, $GHG_{E,t}$ In order to increase greenhouse gas emissions, $LK_t$. For leakage, $\Delta C_{BSL,t}$ is the change in baseline carbon stock, t specific years.

From the current situation of forestry carbon sink development in China, although the carbon storage of soil organic carbon is huge, its carbon sink amount will not fluctuate too much like litter and dead wood, and due to the lack of technical research on the calculation method of the above carbon layer, we can not consider measuring these carbon pools, but only measure the aboveground and underground biomass carbon pools. The calculation method of carbon sink variation of the whole project is shown in formula (6):

$$\Delta C_{p,t} = \sum_{i=1}^{I} \sum_{j=1}^{J} \sum_{k=1}^{K} \left( \Delta C_{P\_,AA,ijk,t} + \Delta C_{P\_,BB,ijk,t} \right) \tag{6}$$

Among $\Delta C_{p,t}$ is the change in the carbon stock of the project, $\Delta C_{P\_,AA,ijk,t}$ is the change amount of the aboveground biomass carbon storage, $\Delta C_{P\_,BB,ijk,t}$ is the change of underground biomass carbon storage, t is the specific number of years (a), I is the project carbon layer (I = 1,2 …,) J is a tree species (J = 1,2 … J)

K is the age (a) and l is the baseline carbon layer (l = 1, 2 …)

The carbon storage calculation formulas of aboveground and underground biomass carbon pools are distributed as follows:

$$\Delta C_{P_{AA},ijk,t} = \Delta C_{P_{Tr},AA,i,k,j,t} + \Delta C_{P\_S,AA,i,k,j,t} \tag{7}$$

$$\Delta C_{P_{BB},ijk,t} = \Delta C_{P\_Tr,BB,i,k,j,t} + \Delta C_{P\_S,BB,i,k,j,t} \tag{8}$$

$\Delta C_{P\_Tr,AA,i,k,j,t}$ is the variation of aboveground biomass carbon storage of the stand.

$\Delta C_{P\_S,AA,i,k,j,t}$ is the change in physical carbon stock.

$\Delta C_{P\_Tr,BB,i,k,j,t}$ is the change amount of underground biomass carbon storage of forest stand.

$\Delta C_{P\_S,BB,i,k,j,t}$ is the change of underground biomass carbon storage of shrub species.

Aboveground biomass is the most important carbon pool in the development of forestry carbon sink, and there are many ways to measure and predict biomass. This study estimates the aboveground biomass of forest trees based on BP neural network. The estimated biomass of the tree is converted into carbon content, and then the carbon content is converted into the equivalent of the carbon dioxide in the atmosphere finally absorbed by the tree, according to the following formula:

$$\Delta C_{P\_Tr,AA,ijk,t} = B_{P\_Tr,ijk,t} * CF_j * A_{ijk} \tag{9}$$

$$\Delta C_{P\_Tr,AA,ijk,t} = \frac{44}{12}(C_{P\_Tr,AA,ijk,t} - C_{P\_Tr,AA,ijk,t-1}) \tag{10}$$

Among $C_{P\_Tr,AA,ijk,t}$ Is the carbon storage in the aboveground biomass carbon pool of the stand.

$B_{P\_Tr,ijk,t}$ Biomass of tree species on the ground of the stand.

$CF_j$ is the average carbon content.

$A_{ijk}$ is the area of the stand (ha).

44/12 is the molecular weight ratio of $CO_2$ to C.

The underground biomass carbon pool is also a crucial carbon pool in the forestry carbon sequestration project. The biomass estimated by the model is combined with formula (11) and formula (12) to calculate the final carbon storage in the underground biomass carbon pool, namely:

$$\Delta C_{P\_Tr,BB,ijk,t} = B_{P\_Tr,ijk,t} * CF_j * A_{ijk} \tag{11}$$

$$\Delta C_{P\_Tr,BB,ijk,t} = \frac{44}{12}\left(C_{P\_Tr,BB,ijk,t} - C_{P\_Tr,BB,ijk,t-1}\right) \tag{12}$$

Among $\Delta C_{P\_Tr,BB,ijk,t}$ is the carbon storage in the below-ground biomass carbon pool of the stand.

$C_{P\_Tr,BB,ijk,t-1}$ stand below-ground biomass carbon storage when is year.

$B_{P\_Tr,ijk,t}$ is the biomass of underground tree species J in the stand, t.

$CF_j$ is the average carbon content (no unit).

$A_{ijk}$ Is the area of the stand (ha).

The carbon storage in the shrub biomass carbon pool can be obtained by converting the biomass results calculated above, namely:

$$\Delta C_{P\_S,AA,ijk,t} = \frac{44}{12}\Delta B_{P\_S,AA,ijk,t} * CF_j * A_{ijk} \tag{13}$$

$$\Delta C_{P\_S,BB,ijk,t} = \frac{44}{12}\Delta B_{P\_S,BB,ijk,t} * CF_j * A_{ijk} \tag{14}$$

$\Delta C_{P\_S,AA,ijk,t}$ Is the change amount of the carbon storage in the biomass carbon pool on the bamboo forest or the Bush forest land

$\triangle C_{P\_S,BB,ijk,t}$ Is the change amount of the carbon storage in the underground biomass carbon pool of the bamboo forest or shrub forest

$\triangle B_{P\_S,AA,ijk,t}$ It is the change of aboveground biomass per unit area of bamboo forest or shrub forest.

$\triangle B_{P\_S,BB,ijk,t}$ Underground biomass per unit area of bamboo forest or shrub forest

$CF_j$ Average carbon content rate of bamboo forest or shrub forest

$A_{ijk}$ Is the area of bamboo forest or shrubbery (hm2)

44/12 is the molecular weight ratio of $CO_2$ to C

T is the specific number of years (a)

I is the project carbon layer

J is the tree species

K is age (a)

For the parameters involved in the above formula, the corresponding default values can be found in the forestry literature, and the shrub parameters of the same species from the local region are preferred when selecting.

It is necessary to consider the choice of greenhouse gas emissions in forestry carbon sequestration projects, and the main emission source is $CO_2$ emissions caused by machinery related to afforestation activities. Increases in greenhouse gas emissions are estimated as follows:

$$GHG_{E,t} = E_{q,t} + E_{N_{Fertilizer},t} \tag{15}$$

$GHG_{E,t}$ Is the increase in greenhouse gas emissions

$E_{q,t}$ Is the increase in CO2 caused by the fuel machine

$E_{N_{Fertilizer},t}$ O2 emissions due to the use of fertilizer

T is the specific number of years (a)

According to the actual development conditions of forestry carbon sink in China, the transfer of forestry activities in advance is not considered. On the other hand, the $CO_2$ emissions generated by tools have been calculated in the greenhouse gas emissions. Therefore, after considering the above issues, the leakage of afforestation activities can be simplified as 0, that is, = 0, where is the leakage generated by the project. For all forestry carbon sequestration open projects, it is necessary to establish their baseline scenarios. In order to calculate the amount of carbon sequestration more accurately after the implementation of the project, it is necessary to calculate the baseline carbon sequestration in advance.

Because the forestry carbon sequestration development project is a new afforestation activity, the newly developed project generally needs to identify whether there are production activities before the project development, if there are no production activities. A portion of the carbon pool can be ignored. It is only necessary to consider the aboveground and underground biomass carbon storage. Namely:

$$\triangle C_{BSL,t} = \sum_{i=1}^{I} (\triangle C_{BSL,AA,i,t} + \triangle C_{BSL,BB,i,t}) \tag{16}$$

$\triangle C_{BSL,t}$ Is the change in baseline carbon stock

$\triangle C_{BSL,AA,i,t}$ Changes in aboveground biomass carbon storage in the baseline carbon layer

$\triangle C_{BSL,BB,i,t}$ Changes in below-ground biomass carbon storage in the baseline carbon horizon

I is the total number of baseline carbon layers

T is the specific number of years (a)

In the baseline scenario, because the project site will be cleared in advance, there is no miscellaneous wood before the development of some projects, and the change of the baseline carbon layer can be regarded as 0.

## Verification and analysis of carbon sink measurement model

The growth of trees has a certain regularity, so the key factors such as tree diameter at breast height and tree height in the process of tree growth have strong time series characteristics and autocorrelation. Therefore, the carbon sequestration model proposed in this paper can be used to predict the forest carbon sequestration, and the reliability of the carbon sequestration model based on biomass method is analyzed through the results.

The data source for the prediction of forest carbon sink is the data from the Forestry Department of Guizhou Province and the ecological products trading platform of Guizhou Province. The selection of growth period of Chinese fir should be considered in the model validation measurement, because there will be over-mature forest and young forest with insufficient carbon sink potential in the whole forest land, so the over-mature forest and young forest should be discarded in the data preprocessing process. Finally, the corresponding results were obtained according to the average DBH and average height of the treated Cunninghamia lanceolata at different ages. The prediction results are shown in the Table 2.

It can be seen from the data analysis in the table that the prediction result is relatively ideal, the average relative error is 6.09%, and the absolute error range is 0.348–4.262/hm 2. The results again show that the biomass estimation model and carbon sequestration measurement model based on BP neural network established in this paper have high accuracy and reliability.

## Calculation and model validation of carbon sequestration in field monitoring area

In order to verify the accuracy of the model, we calculated the data of two sample plots monitored in the field of ecological restoration and compared them with the predicted data of the model.

## Basic data collection

The data in this paper mainly come from the combination of field survey and data collection, through the field survey to obtain the tree height, diameter at breast height, crown width, height under branch, phenological period and viability of the arbors in the sample plot, and through the statistical yearbook, local chronicles and relevant literature of the project area to collect the local natural environment, natural resources, social and economic conditions and other information.

In this paper, the monitoring study plots are located in Majiatian Coal Mine and fei Longhu Wetland Park(Yu Qin), and two fixed plots are established for long-term monitoring. The construction and division of sample plots shall be carried out according to the technical specifications of the Center for Tropical Forest Science.

At the same time of establishing the sample plot, each tree species was numbered in the 10 m × 10 m sample plot. The number of the tree species was determined by the row number and column number of the tree species in the sample plot. The lower left corner of the sample plot was taken as the origin to arrange the column number and row number in the sample plot in the order of the coordinate axis. It is worth noting that the number first indicates the position of the column, and then indicates the position of the row. The individual numbering method of the tree species is as follows: firstly, English letters representing the location of the sample plot and Roman numerals representing the serial number of the waste dump are used together to represent the precise location of the sample plot, four-digit Arabic numerals are used to represent the location of the small sample in the sample plot, and a three-digit Arabic numeral is used to represent the specific tree number of the tree species.

Table 2. Forestland carbon storage prediction results.

| Planting age (years) | Mean diameter at breast height | Average plant height | Predicted value | Observation value | Absolute error | Relative error |
|---|---|---|---|---|---|---|
| | (cm) | (m) | (t/hm2) | (t/hm2) | (t/hm2) | (%) |
| 5 | 8.75 | 5.37 | 19.559 | 21.099 | 1.54 | 7.30 |
| 6 | 9.25 | 5.97 | 21.667 | 24.282 | 2.615 | 10.77 |
| 7 | 9.75 | 6.87 | 24.749 | 29.011 | 4.262 | 14.69 |
| 8 | 10.35 | 7.87 | 28.323 | 31.023 | 2.7 | 8.70 |
| 9 | 11.15 | 8.27 | 31.762 | 33.045 | 1.283 | 3.88 |
| 10 | 11.65 | 8.97 | 33.591 | 37.931 | 4.34 | 11.44 |
| 11 | 12.15 | 9.87 | 36.897 | 39.971 | 3.074 | 7.69 |
| 12 | 12.65 | 10.07 | 40.757 | 41.54 | 0.783 | 1.88 |
| 13 | 13.15 | 10.67 | 42.778 | 45.714 | 2.936 | 6.42 |
| 15 | 13.65 | 11.07 | 44.295 | 45.934 | 1.639 | 3.57 |
| 16 | 14.15 | 11.77 | 46.825 | 48.803 | 1.978 | 4.05 |
| 17 | 14.35 | 11.97 | 47.902 | 51.861 | 3.959 | 7.63 |
| 18 | 14.75 | 12.87 | 50.307 | 52.017 | 1.71 | 3.29 |
| 19 | 14.95 | 12.87 | 52.97 | 53.877 | 0.907 | 1.68 |
| 20 | 15.15 | 13.27 | 52.081 | 56.269 | 4.188 | 7.44 |
| 21 | 15.45 | 13.57 | 54.735 | 56.141 | 1.406 | 2.50 |
| 22 | 15.75 | 13.97 | 57.632 | 57.98 | 0.348 | 0.60 |

In the 10 m × 10 m quadrats, the individuals with the diameter at breast height (DBH) ≥ 1 cm were marked clockwise from the lower left corner. For trees with branches, the branches should also be labeled, for individual trunks or branches with DBH less than 5 cm, the label should be fixed with fishing line, for individual trees with DBH ≥ 5 cm, the label should be fixed with stainless steel nails, the height of the label from the ground is about 1.5 m, and the angle of the steel nails is about 60 degrees with the trunk obliquely upward. The depth of nailing into the trunk is about 1 cm, and the stumps for various reasons are marked. This verification only includes Majiatian Coal Mine (No.1 sample plot) and Yuqingfei Longhu Wetland Park (No.2 sample plot).

The DBH of tree species with DBH less than 5 cm shall be measured with vernier caliper, and the DBH of tree species with DBH greater than 5 cm shall be measured with DBH ruler; if there are branches below the DBH of tree species and the DBH of branches is greater than 1 cm, the DBH of branches shall be measured; When the tree height of some tree species in the sample plot is less than 1. 3 m, the diameter at breast height at 1. 3 m cannot be measured, so the vernier caliper is used to measure the base diameter, and the steel tape is used to measure and record the coordinate position in the 5 m × 5 m small quadrat. In the process of field survey, besides measuring the diameter at breast height of trees, it is also necessary to accurately record the name of the investigated tree species, tree number (tag), species name (species), coordinates (GX), tree height, crown width, phenological period and vitality, and to establish a sample tree species database.

The input of survey data adopts the dual-track method of letting two people input the same data, and checks the input results to ensure the correctness of data input. Use the relevant software Excel compare to compare the input results of the two people, the software will prompt the difference between the two input results, and then check the data to correct the wrong data. In order to facilitate the sharing of sample survey data and the processing of data, the database is standardized according to the standard requirements of CTFS, and the standardized database is Excel file.

## Standing wood volume analysis of each plot

The total volume in plot 1 was 270 m³, and the volume density was 32. 61 m³· hm ⁻². The difference between the total volume of surviving individuals and the total volume of trees in the plot was only 0. 67 m³, the proportion of the total volume of surviving individuals to the total volume of trees was 97. 26%, and the density of surviving volume was 29. 68 m³· hm⁻².

The purpose of data unification is to unify the format and name fields of the data to be analyzed, such as the unification of different measurement units and species information. For the unification of measurement units, the professional forestry involved in this study, such as the three parameters of biomass, diameter at breast height and tree height, are measured in kg, trees per hectare, initial planting density per hectare, scattered tree species, growth and other survey factors. Human activity had little impact in the selected study area and growth was stable over time. Exclude special circumstances such as forestry activities, fruit tree planting and being in the urban green belt in the study area. Finally, the collected data is divided into two parts, which are used for training and simulation.(Table 3,Table 4)

The total volume in plot 2 was 36. 00 m³, and the volume density was 45. 00 m³· hm ⁻². The total volume of surviving trees was 30. 30 m³, which was 5. 70 m higher than total volume of trees³, and the density of surviving trees was 37. 88 m³· hm ⁻². The volume data in the sample plot is shown in the following Table 5, Table 6.

Forest stock volume refers to the total volume of existing trees in a certain area of forest. It generally refers to the volume of the trunk including bark. Sometimes, in order to get more accurate results, it is calculated separately according to tree species, diameter class, timber species, etc. The unit of forest stock volume is cubic meter. In this paper, the forest volume is calculated separately according to the tree species in the same plot. The final calculation results are shown in the following table, and the total volume of each tree species and the average volume of a single tree are compared and analyzed.(Table 7)

Based on the measured data, the results showed that the biomass of plot 1 was the highest, reaching 55. 08 t (for the convenience of comparison, the area of plot was calculated as 1. 0 hm², the same below). The carbon sink density is 21.24 t· hm ⁻². The carbon sink of plot 2 was 21. 65 t, and the biomass of plot 1 was 56. 58 t, and the carbon sink of plot 2 was 22. 26 t. The results show that the average relative error is less than 2.5% and the absolute error ranges from 0.61 to 1.5t/hm², which verifies the reliability of the model.

**Table 3. Volume of Standing Timber in No.1 Sample Plot.**

| Sample group number | Timber volume of Robinia pseudoacacia | Number of Robinia pseudoacacia | Volume of Pinus tabulaeformis | Number of Pinus tabulaeformis | Volume of elm trees | Number of elm trees |
|---|---|---|---|---|---|---|
| 1 | 3.82 | 156 | 0.31 | 38 | 5.54 | 19 |
| 2 | 1.82 | 121 | 0.23 | 36 | 5.72 | 19 |
| 3 | 2.41 | 122 | 0.26 | 38 | 8.29 | 21 |
| 4 | 2.43 | 96 | 0.19 | 32 | 2.78 | 19 |
| 5 | 2.31 | 86 | 0.13 | 36 | 19.98 | 24 |
| 6 | 2.51 | 95 | 0.2 | 35 | 6.46 | 19 |
| 7 | 2.46 | 103 | 0.18 | 33 | 24.94 | 23 |
| 8 | 1.82 | 90 | 0.22 | 37 | 17.09 | 26 |
| 9 | 2.21 | 105 | 0.23 | 34 | 7.94 | 22 |
| 10 | 1.35 | 62 | 0.19 | 34 | 12.92 | 17 |

**Table 4.  Table of Living Wood Volume of No.1 Sample Plot.**

| Sample group number | Timber volume of Robinia pseudoacacia | Number of Robinia pseudoacacia | Volume of Pinus tabulaeformis | Number of Pinus tabulaeformis |
|---|---|---|---|---|
| 1 | 4.91 | 71 | 0.46 | 34 |
| 2 | 3.08 | 71 | 0.38 | 32 |
| 3 | 3.25 | 70 | 0.41 | 34 |
| 4 | 3.61 | 74 | 0.32 | 28 |
| 5 | 3.69 | 85 | 0.28 | 32 |
| 6 | 3.31 | 74 | 0.35 | 31 |
| 7 | 3.58 | 73 | 0.33 | 29 |
| 8 | 2.87 | 69 | 0.37 | 32 |
| 9 | 3.38 | 72 | 0.38 | 30 |
| 10 | 2.43 | 53 | 0.34 | 30 |

**Table 5.  Volume of Standing Timber in No.2 Sample Plot.**

| Sample group number | Timber volume of Robinia pseudoacacia (m3) | Number of Robinia pseudoacacia | Volume of Pinus tabulaeformis (m3) | Number of Pinus tabulaeformis | Volume of elm trees (m3) | Number of elm trees |
|---|---|---|---|---|---|---|
| 1 | 3.63 | 128 | 0.25 | 23 | 0.59 | 51 |
| 2 | 3.45 | 111 | 0.22 | 15 | 0.86 | 43 |
| 3 | 2.45 | 95 | 0.36 | 19 | 0.61 | 29 |
| 4 | 3.06 | 120 | 0.18 | 16 | 0.33 | 39 |
| 5 | 3.38 | 136 | 0.11 | 13 | 0.11 | 37 |
| 6 | 3.04 | 129 | 0.23 | 16 | 0.40 | 49 |
| 7 | 3.10 | 121 | 0.25 | 21 | 0.34 | 47 |
| 8 | 2.99 | 121 | 0.18 | 16 | 0.24 | 34 |
| 9 | 3.10 | 130 | 0.30 | 16 | 0.14 | 49 |
| 10 | 2.50 | 124 | 0.24 | 15 | 0.56 | 45 |

**Table 6.  Table of Living Wood Volume of No.2 Sample Plot.**

| Sample group number | Timber volume of Robinia pseudoacacia (m3) | Number of Robinia pseudoacacia | Volume of Pinus tabulaeformis (m3) | Number of Pinus tabulaeformis |
|---|---|---|---|---|
| 1 | 3.51 | 92 | 0.24 | 20 |
| 2 | 2.61 | 72 | 0.19 | 14 |
| 3 | 2.45 | 95 | 0.36 | 18 |
| 4 | 2.89 | 83 | 0.17 | 15 |
| 5 | 3.22 | 81 | 0.10 | 11 |
| 6 | 2.52 | 74 | 0.23 | 16 |
| 7 | 2.94 | 86 | 0.25 | 20 |
| 8 | 2.82 | 77 | 0.18 | 16 |
| 9 | 2.92 | 91 | 0.29 | 16 |
| 10 | 2.16 | 71 | 0.23 | 15 |
| Subtotal | 28.05 | 822 | 2.25 | 161 |

## Conclusion

Since the 1990s, there have been many studies on forest biomass abroad. Ter-Mi-kaelian & korzukhin [28]reviewed the modeling equations for standing tree biomass of 65 tree species in North America. A total of 803 modeling equations of the form $M = aD^b$ were given. Parresol [29] reviewed the biomass models and the methods of weighted analysis, discussed not only the problems of additivity and coordination, the error components of stand biomass estimation, but also the evaluation of biomass model indices. Biomass equations of six tree species in the northern forests of Manitoba, Canada, were established by Bond Lamberty et al [30].Snorrason & Einarsson [31] established the stem volume equation and aboveground biomass equation of 11 main tree species in Iceland. The total above-ground volume of seven main tree species in France was established Vallet et al. [32] Biomass models were developed Basuki et al [33]for four Dipterocarpaceae genera in forests in the eastern region of Galinmandan, Indonesia. N Návar developed the relative growth equations for foliage, trunk, biomass, and stem biomass for 10 tree species in the temperate and tropical forests of northwestern Mexico. The study of forest biomass in China started late [34–41]. Since the 1990s, the biomass of Pinus massoniana in southern China has been studied, and the compatible tree biomass model compatible with volume has been established by the method of linear simultaneous equations. Taking Cunning-hamia lanceolata in southern China as the research object, Zeng Weisheng et al established a nonlinear biomass model system, which not only considered the compatibility of biomass and volume, but also satisfied that the total biomass of trees was equal to the sum of all components. Xu Hui studied the biomass of Larix gmelinii, Tilia amurensis and Abies faxoniana in northern China, and established the models of biomass of each part and total biomass compatible with volume. Zhang Huiru studied the biomass of Tilia amurensis and Larix gmelinii, and established the nonlinear models of total biomass and biomass of each part with volume, diameter, tree height, crown width, crown length and other factors. Taking Larix gmelinii in Northeast China as the research object, Tang Shouzheng made a comparative analysis and evaluation of three biomass estimation methods and two proportional adjustment methods. Different experimental results show that the prediction of belowground biomass and aboveground biomass is very close to the expected results, but the prediction of belowground biomass is still not ideal [42–47]. Biomass evaluation index pairs of different models are shown in Table 8 and Table 9.

Through the study, it is found that compared with the traditional binary regression model, the neural network is more accurate in estimating the determination coefficient R2 of biomass, while the MAE, RMSE and MSE index coefficients are getting smaller and smaller, indicating that the model fitting accuracy is better.

It can be seen from the results that all the indexes of the optimized neural network are close to the better parameters. The prediction accuracy of the model for the underground biomass of the tree species is greatly improved.

This paper mainly deduces an accurate and feasible carbon sequestration model based on biomass method. This paper first introduces the concept of carbon sink project boundary

**Table 7. Forest Volume in Each Plot.**

| Sample plot information | Plot 1 | Plot 2 |
|---|---|---|
| Forest stock | | |
| (m3) | 24.41 | 36.00 |
| Volume of surviving wood | | |
| (m3) | 23.74 | 30.30 |
| Stand volume density | | |
| (m3/hm2) | 30.51 | 45.00 |

**Table 8. Comparison of aboveground biomass evaluation indexes of different models.**

| Model | Evaluation index | | | |
|---|---|---|---|---|
| | R2 | MAE | RMSE | MSE |
| Traditional binary regression model | 0.8102 | – | – | – |
| BP neural network model | 0.927 | 15.6935 | 24.8975 | 619.8871 |
| Neural Network Model after Replacing sigmoid Function with reLU | 0.9702 | 9.2669 | 15.9194 | 253.4283 |
| Neural network model after replacing MSEloss with binary _ crossentropy; | 0.9829 | 6.5266 | 12.0536 | 145.2886 |

**Table 9. Comparison of Underground Biomass Evaluation Indexes of Different Models.**

| Model | Evaluation index | | | |
|---|---|---|---|---|
| | R2 | MAE | RMSE | MSE |
| Traditional binary regression model | 0.8102 | – | – | – |
| BP neural network model | 0.927 | 15.6935 | 24.8975 | 619.8871 |
| Neural Network Model after Replacing sigmoid Function with reLU | 0.9702 | 9.2669 | 15.9194 | 253.4283 |
| Neural network model after replacing MSEloss with binary _ crossentropy; | 0.9829 | 6.5266 | 12.0536 | 145.2886 |

and how to determine the project boundary, and also introduces the selection and decision-making methods of carbon pool and greenhouse gas emissions. In the process of measurement, considering that it is impossible to calculate all the forest land, this simulation derives the corresponding sample plot sampling calculation formula and designs the sample plot on the premise of ensuring the accuracy requirements. Then the whole carbon sequestration model is constructed, in which sampling design, project boundary, carbon pool selection and other factors are considered comprehensively, and the actual net carbon sequestration caused by $CH_4$, $N_2O$ and other gas emissions in forestry carbon sequestration projects is not accurate. Combined with the actual project development needs, the specific calculation model is derived. Finally, according to the strong time series characteristics of key factors such as diameter at breast height and tree height in the process of forest growth, taking Cunninghamia lanceolata as an example, the forestry carbon sequestration measurement model in this chapter was verified by using this characteristic, the average relative error was 6.09%, and the absolute error range was 0.348–4.262/hm², thus the reliability of the model was verified.

## Author contributions

**Data curation:** Qiu Yan.

**Investigation:** Li Chunling.

**Methodology:** Li Chunling.

**Project administration:** Yan Jun.

**Resources:** Yan Jun, Ran Wen Rui.

**Validation:** Ran Wen Rui.

**Writing – original draft:** Zhang Min.

**Writing – review & editing:** Wu Yang.

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
