## [Decision Letter · Decision Letter 0]

16 Sep 2024

PONE-D-24-32522Research and Validation of Forest Carbon Sequestration Measurement Model Based on Biomass Method-A Case Study of Guizhou ProvincePLOS ONE

Dear Dr. Yang,

Thank you for submitting your manuscript to PLOS ONE. After careful consideration, we feel that it has merit but does not fully meet PLOS ONE’s publication criteria as it currently stands. Therefore, we invite you to submit a revised version of the manuscript that addresses the points raised during the review process.

We look forward to receiving your revised manuscript.

Kind regards,

Dafeng Hui, Ph.D.

Academic Editor

PLOS ONE

**Journal Requirements:**

This research was supported by the Concealed Ore Deposit Exploration and Innovation Team of Guizhou Colleges and Universities (Guizhou Education and Cooperation Talent Team [2015] 56), Provincial Key Discipline of Geological Resources and Geological Engineering of Guizhou Province (ZDXK [2018] 001), Huang Danian Resources of the National Colleges and Universities Teachers' Team of Exploration Engineering (Teacher Letter [2018] No. 1), Geological Resources and Geological Engineering Talent Base of Guizhou Province (RCJD2018-3), Key Laboratory of Karst Engineering Geology and Hidden Mineral Resources of Guizhou Province (Qianjiaohe KY [2018] No. 486 Guizhou Institute of Technology Rural Revitalization Soft Science Project [2022xczx10]), and the Education and Teaching Reform Research Project of Guizhou Institute of Technology (grant numbers JGZD202107 and 2022TDFJG01).

the sponsors or funders do not play any role in the study design, data collection and analysis, decision to publish, or preparation of the manuscript

**Additional Editor Comments:**

I now have two reports on this manuscript. Both reviewers recognized the importance of the study, but provide two different recommendations. Reviewer #2 was quite positive and suggested adding more details on the methods. Reviewer #1 was more critical, and questioned the technique issues of the manuscript, particularly the validation of the carbon sink measurement model and a lack of sufficient discussion of the results. Based on their evaluations, my recommendation is Major revision. The authors need to carefully address the issues raised by the reviewers and make a substantial revision of the manuscript.

Reviewers' comments:

Reviewer's Responses to Questions

**Comments to the Author**

1. Is the manuscript technically sound, and do the data support the conclusions?

Reviewer #1: Partly

Reviewer #2: Yes

2. Has the statistical analysis been performed appropriately and rigorously? 

Reviewer #1: No

Reviewer #2: Yes

3. Have the authors made all data underlying the findings in their manuscript fully available?

Reviewer #1: Yes

Reviewer #2: Yes

4. Is the manuscript presented in an intelligible fashion and written in standard English?

Reviewer #1: Yes

Reviewer #2: Yes

5. Review Comments to the Author

**Reviewer #1:**  Title of Manuscript: Research and Validation of Forest Carbon Sequestration Measurement Model Based on Biomass Method-A Case Study of Guizhou Province

Comments to Editor:

The study takes into account the urgent need for carbon sequestration measurement in Guizhou Province of China, which has a large potential for carbon sequestration, after the establishment of China's Certified Voluntary Emission Reduction (CCER) market, which is a research work of some value and significance. However, the current article is weak in overall research compendium, and the current research progress is described vaguely and lacks depth and breadth. At the same time, the article also has major structural defects that need to be improved, and the validation of the carbon sink measurement model is not persuasive enough and lacks sufficient discussion, which makes it difficult to convince people of the usability of the model. The specific comments are as follows:

Comments to Author:

1.As a methodological study, model accuracy and transferability are extremely important, but the validation of this paper is relatively simple, only using a set of data(Table 2), which is not representative and illustrative of the performance of the model in this paper, and it is recommended that one or more regions be selected as a case area for validation.

2.This paper lacks a discussion of the results of the derivation of the formula, which I believe is a major flaw, a valuable literature should be discussed in depth in the discussion section to discuss the contribution of this study, and the results of previous research results are compared to illustrate the strengths and weaknesses of the results of the study.

3.Lines 50-55: Lack of source references for data listed in the introduction.

4.Lines 57-58: In the introduction, it is mentioned that some traditional research methods have the problems of heavy workload and too much reliance on manual labor, which research methods are specifically referred to, and please give examples of traditional methods in the existing literature to provide an in-depth compendium and summarize the progress.

5.Line 76: Is the capital letter “O” a misprint? Or does it have a specific meaning? This is difficult to understand, so please clarify.

6.Lines 236-240, the topic of the article as well as the entire text talks about forest carbon sinks, so why does the sampling design jump to agricultural carbon sinks? This is a basic logical flaw.

7.Lines 435-437 state that this paper derives a formula for sample plot sampling, which, as far as I know, is derived from the methodology for afforestation carbon sink projects (CCER-14-001-V01), what is the difference between the two? Please clarify.

8.the text of a large part of the reference afforestation carbon sink project methodology (CCER-14-001-V01) to do a lot of qualitative description, if as a carbon sink development project it is necessary, but for a scientific research theme, I think that padding is more redundant.

9.The article has fewer references (only 16), which is a side reflection of the inadequacy of this paper to sort out the current status and progress of research.

**Reviewer #2: ** The author tries to use the BP neural network prediction method to estimate the forest biomass quickly. The input variables of this method take into account the error of forest growth data caused by different geographical locations. Then, based on the biomass estimation model, the author studies the construction of carbon sequestration measurement model. It can be seen that a lot of work has been done and some conclusions have been drawn, which is very interesting. I think the article still has the following problems.

1. At present, there is no clear specification for the measurement method of forest carbon sequestration. The author chooses the biomass method, and its accuracy and principle need to be explained in the text.

2. There is a forestry carbon sequestration measurement and prediction system or platform mentioned in the article, but it is not actually pointed out. The author needs to add a case to illustrate that forestry carbon sequestration has been recognized to some extent.

3. Under the baseline scenario, the baseline carbon layer change can be regarded as 0. Why? 4. For all forestry carbon sinks, the aboveground biomass carbon pool is the most important. How did the author select this part?

4. Most of the sample plots set by the project are rectangular or square, but the forest on the spot can not be so regular, how to eliminate such errors?

6. The English expression of the article needs to be strengthened My suggestion is minor revision. Once the author can complete the above questions, I think the article can be accepted.

6. PLOS authors have the option to publish the peer review history of their article (what does this mean? ). If published, this will include your full peer review and any attached files.

**Do you want your identity to be public for this peer review?** For information about this choice, including consent withdrawal, please see our Privacy Policy .

Reviewer #1: No

Reviewer #2: No

---

## [Author Response · Author response to Decision Letter 1]

15 Oct 2024

Dear editor

Thank you very much for your letter. We have learned much from your and two reviewers’ comments, which are fair, encouraging and constructive. After carefully studying the comments and your advice, we have made corresponding changes. The main revisions are listed below.

For reviewer one:

1.At present, there is no clear specification for the measurement method of forest carbon sequestration. The author chooses the biomass method, and its accuracy and principle need to be explained in the text.

Thanks to the opinions of the reviewer, This suggestion is very good and scientific. Indeed, we need to supplement the field data to verify the accuracy of the model. We have several field observation points in the field. I added the measured data of two points to verify the model. The article also added a corresponding chapter: Calculation and Model Validation of Carbon Sequestration in Field Monitoring Area.

Calculation and Model Validation of Carbon Sequestration in Field Monitoring Area

In order to verify the accuracy of the model, we calculated the data of two sample plots monitored in the field of ecological restoration and compared them with the predicted data of the model.

Basic data collection

The data in this paper mainly come from the combination of field survey and data collection, through the field survey to obtain the tree height, diameter at breast height, crown width, height under branch, phenological period and viability of the arbors in the sample plot, and through the statistical yearbook, local chronicles and relevant literature of the project area to collect the local natural environment, natural resources, social and economic conditions and other information.

In this paper, the monitoring study plots are located in Majiatian Coal Mine and fei Longhu Wetland Park(Yu Qin), and two fixed plots are established for long-term monitoring. The construction and division of sample plots shall be carried out according to the technical specifications of the Center for Tropical Forest Science.

At the same time of establishing the sample plot, each tree species was numbered in the 10 m × 10 m sample plot. The number of the tree species was determined by the row number and column number of the tree species in the sample plot. The lower left corner of the sample plot was taken as the origin to arrange the column number and row number in the sample plot in the order of the coordinate axis. It is worth noting that the number first indicates the position of the column, and then indicates the position of the row. The individual numbering method of the tree species is as follows: firstly, English letters representing the location of the sample plot and Roman numerals representing the serial number of the waste dump are used together to represent the precise location of the sample plot, four-digit Arabic numerals are used to represent the location of the small sample in the sample plot, and a three-digit Arabic numeral is used to represent the specific tree number of the tree species.

In the 10 m×10 m quadrats, the individuals with the diameter at breast height (DBH) ≥ 1 cm were marked clockwise from the lower left corner. For trees with branches, the branches should also be labeled, for individual trunks or branches with DBH less than 5 cm, the label should be fixed with fishing line, for individual trees with DBH ≥ 5 cm, the label should be fixed with stainless steel nails, the height of the label from the ground is about 1.5 m, and the angle of the steel nails is about 60 degrees with the trunk obliquely upward. The depth of nailing into the trunk is about 1 cm, and the stumps for various reasons are marked. This verification only includes Majiatian Coal Mine (No.1 sample plot) and Yuqingfei Longhu Wetland Park (No.2 sample plot).

The DBH of tree species with DBH less than 5 cm shall be measured with vernier caliper, and the DBH of tree species with DBH greater than 5 cm shall be measured with DBH ruler; if there are branches below the DBH of tree species and the DBH of branches is greater than 1 cm, the DBH of branches shall be measured; When the tree height of some tree species in the sample plot is less than 1. 3 m, the diameter at breast height at 1. 3 m cannot be measured, so the vernier caliper is used to measure the base diameter, and the steel tape is used to measure and record the coordinate position in the 5 m × 5 m small quadrat. In the process of field survey, besides measuring the diameter at breast height of trees, it is also necessary to accurately record the name of the investigated tree species, tree number (tag), species name (species), coordinates (GX), tree height, crown width, phenological period and vitality, and to establish a sample tree species database.

The input of survey data adopts the dual-track method of letting two people input the same data, and checks the input results to ensure the correctness of data input. Use the relevant software Excel compare to compare the input results of the two people, the software will prompt the difference between the two input results, and then check the data to correct the wrong data. In order to facilitate the sharing of sample survey data and the processing of data, the database is standardized according to the standard requirements of CTFS, and the standardized database is Excel file.

Standing wood volume analysis of each plot

The total volume in plot 1 was 270 m3 , and the volume density was 32. 61 m3 · hm -2 . The difference between the total volume of surviving individuals and the total volume of trees in the plot was only 0. 67 m3 , the proportion of the total volume of surviving individuals to the total volume of trees was 97. 26%, and the density of surviving volume was 29. 68 m3 · hm-2 .

The purpose of data unification is to unify the format and name fields of the data to be analyzed, such as the unification of different measurement units and species information. For the unification of measurement units, the professional forestry involved in this study, such as the three parameters of biomass, diameter at breast height and tree height, are measured in kg, trees per hectare, initial planting density per hectare, scattered tree species, growth and other survey factors. Human activity had little impact in the selected study area and growth was stable over time. Exclude special circumstances such as forestry activities, fruit tree planting and being in the urban green belt in the study area. Finally, the collected data is divided into two parts, which are used for training and simulation.(Table 3-4)

Table 3 Volume of Standing Timber in No.1 Sample Plot

Sample group number Timber volume of Robinia pseudoacacia Number of Robinia pseudoacacia Volume of Pinus tabulaeformis Number of Pinus tabulaeformis Volume of elm trees Number of elm trees

1 3.82 156 0.31 38 5.54 19

2 1.82 121 0.23 36 5.72 19

3 2.41 122 0.26 38 8.29 21

4 2.43 96 0.19 32 2.78 19

5 2.31 86 0.13 36 19.98 24

6 2.51 95 0.2 35 6.46 19

7 2.46 103 0.18 33 24.94 23

8 1.82 90 0.22 37 17.09 26

9 2.21 105 0.23 34 7.94 22

10 1.35 62 0.19 34 12.92 17

Table 4 Table of Living Wood Volume of No.1 Sample Plot

Sample group number Timber volume of Robinia pseudoacacia Number of Robinia pseudoacacia Volume of Pinus tabulaeformis Number of Pinus tabulaeformis

1 4.91 71 0.46 34

2 3.08 71 0.38 32

3 3.25 70 0.41 34

4 3.61 74 0.32 28

5 3.69 85 0.28 32

6 3.31 74 0.35 31

7 3.58 73 0.33 29

8 2.87 69 0.37 32

9 3.38 72 0.38 30

10 2.43 53 0.34 30

The total volume in plot 2 was 36. 00 m 3 , and the volume density was 45. 00 m 3 · hm -2 . The total volume of surviving trees was 30. 30 m 3 , which was 5. 70 m higher than total volume of trees 3 , and the density of surviving trees was 37. 88 m 3 · hm -2 . The volume data in the sample plot is shown in the following Table5-6.

Sample groupnumber Timber volume of Robinia pseudoacacia

m3� Number of Robinia pseudoacacia Volume of Pinus tabulaeformis

m3� Number of Pinus tabulaeformis Volume of elm trees

m3� Number of elm trees

1 3.63 128 0.25 23 0.59 51

2 3.45 111 0.22 15 0.86 43

3 2.45 95 0.36 19 0.61 29

4 3.06 120 0.18 16 0.33 39

5 3.38 136 0.11 13 0.11 37

6 3.04 129 0.23 16 0.40 49

7 3.10 121 0.25 21 0.34 47

8 2.99 121 0.18 16 0.24 34

9 3.10 130 0.30 16 0.14 49

10 2.50 124 0.24 15 0.56 45

Table 6 Table of Living Wood Volume of No.2 Sample Plot

Sample group number Timber volume of Robinia pseudoacacia

m3

Number of Robinia pseudoacacia Volume of Pinus tabulaeformis

m3

Number of Pinus tabulaeformis

1 3.51 92 0.24 20

2 2.61 72 0.19 14

3 2.45 95 0.36 18

4 2.89 83 0.17 15

5 3.22 81 0.10 11

6 2.52 74 0.23 16

7 2.94 86 0.25 20

8 2.82 77 0.18 16

9 2.92 91 0.29 16

10 2.16 71 0.23 15

Subtotal 28.05 822 2.25 161

Forest stock volume refers to the total volume of existing trees in a certain area of forest. It generally refers to the volume of the trunk including bark. Sometimes, in order to get more accurate results, it is calculated separately according to tree species, diameter class, timber species, etc. The unit of forest stock volume is cubic meter. In this paper, the forest volume is calculated separately according to the tree species in the same plot. The final calculation results are shown in the following table, and the total volume of each tree species and the average volume of a single tree are compared and analyzed.(Table 7)

Table 7 Forest Volume in Each Plot

Sample plot information Plot 1 Plot 2

Forest stock

m3� 24.41 36.00

Volume of surviving wood

m3� 23.74 30.30

Stand volume density

m3/hm2� 30.51 45.00

Based on the measured data, the results showed that the biomass of plot 1 was the highest, reaching 55. 08 t (for the convenience of comparison, the area of plot was calculated as 1. 0 hm 2 , the same below). The carbon sink density is 21.24 t · hm -2 . The carbon sink of plot 2 was 21. 65 t, and the biomass of plot 1 was 56. 58 t, and the carbon sink of plot 2 was 22. 26 t. The results show that the average relative error is less than 2.5% and the absolute error ranges from 0.61 to 1.5t/hm 2 , which verifies the reliability of the model.

2.This paper lacks a discussion of the results of the derivation of the formula, which I believe is a major flaw, a valuable literature should be discussed in depth in the discussion section to discuss the contribution of this study, and the results of previous research results are compared to illustrate the strengths and weaknesses of the results of the study.

Thanks to the opinions of the reviewer, This suggestion is very good and scientific.In the last conclusion, we add the corresponding content in the discussion part.

Since the 1990s, there have been many studies on forest biomass abroad. Ter-Mi-kaelian

& korzukhin [17]reviewed the modeling equations for standing tree biomass of 65 tree species in North America. A total of 803 modeling equations of the form M = aDb were given. Parresol[18] reviewed the biomass models and the methods of weighted analysis, discussed not only the problems of additivity and coordination, the error components of stand biomass estimation, but also the evaluation of biomass model indices. Biomass equations of six tree species in the northern forests of Manitoba, Canada, were established by Bond Lamberty et al[19].Snorrason & Einarsson[20] established the stem volume equation and aboveground biomass equation of 11 main tree species in Iceland. The total above-ground volume of seven main tree species in France was established Vallet et al.[21] Biomass models were developed Basuki et al[22]for four Dipterocarpaceae genera in forests in the eastern region of Galinmandan, Indonesia. N Návar developed the relative growth equations for foliage, trunk, biomass, and stem biomass for 10 tree species in the temperate and tropical forests of northwestern Mexico. The study of forest biomass in China started late[23-30]. Since the 1990s, the biomass of Pinus massoniana in southern China has been studied, and the compatible tree biomass model compatible with volume has been established by the method of linear simultaneous equations. Taking Cunninghamia lanceolata in southern China as the research object, Zeng Weisheng et al established a nonlinear biomass model system, which not only considered the compatibility of biomass and volume, but also satisfied that the total biomass of trees was equal to the sum of all components. Xu Hui studied the biomass of Larix gmelinii, Tilia amurensis and Abies faxoniana in northern China, and established the models of biomass of each part and total biomass compatible with volume. Zhang Huiru studied the biomass of Tilia amurensis and Larix gmelinii, and established the nonlinear models of total biomass and biomass of each part with volume, diameter, tree height, crown width, crown length and other factors. Taking Larix gmelinii in Northeast China as the research object, Tang Shouzheng made a comparative analysis and evaluation of three biomass estimation methods and two proportional adjustment methods. Different experimental results show that the prediction of belowground biomass and aboveground biomass is very close to the expected results, but the prediction of belowground biomass is still not ideal[31-35]. Biomass evaluation index pairs of different models are shown in Table 8 and Table 9.

Table 8 Comparison of aboveground biomass evaluation indexes of different models

Model Evaluation index

R2 MAE RMSE MSE

Traditional binary regression model 0.8102 - - -

BP neural network model 0.927 15.6935 24.8975 619.8871

Neural Network Model after Replacing sigmoid Function with reLU 0.9702 9.2669 15.9194 253.4283

Neural network model after replacing MSEloss with binary _ crossentropy; 0.9829 6.5266 12.0536 145.2886

Through the study, it is found that compared with the traditional binary regression model, the neural network is more accurate in estimating the determination coefficient R2 of biomass, while the MAE, RMSE and MSE index coefficients are getting smaller and smaller, indicating that the model fitting accuracy is better.

Table 9 Comparison of Underground Biomass Evaluation Indexes of Different Models

Model Evaluation index

R2 MAE RMSE MSE

Traditional binary regression model 0.8102 - - -

BP neural network model 0.927 15.6935 24.8975 619.8871

Neural Network Model after Replacing sigmoid Function with reLU 0.9702 9.2669 15.9194 253.4283

Neural network model after replacing MSEloss with binary _ crossentropy; 0.9829 6.5266 12.0536 145.2886

It can be seen from the results that all the indexes of the optimized neural network are close to the better parameters. The prediction accuracy of the model for the underground biomass of the tree species is greatly improved.

This paper mainly deduces an accurate and feasible carbon sequestration model based on biomass method. This paper first introduces the concept of carbon sink project boundary and how to determine the project boundary, and also introduces the selection and decision-making methods of carbon pool and greenhouse gas emissions. In the process of measurement, considering that it is impossible to calculate all the forest land, this simulation derives the corresponding sample plot sampling calculation formula and designs the sample plot on the premise of ensuring the accuracy requirements. Then the whole carbon sequestration model is constructed, in which sampling design, project boundary, carbon pool selection and other factors are considered comprehensively, and the actual net carbon sequestration caused by CH4 , N2O and other gas emissions in forestry carbon sequestration projects is not accurate. Combined with the actual project development needs, the specific calculation model is derived. Finally, according to the strong time series characteristics of key factors such as diameter at breast height and tree height in the process of forest growth, taking Cunninghamia lanceolata as an example, the forestry carbon sequestration measurement model in this chapter was verified by using this characteristic, the average relative error was 6.09%, and the absolute error range was 0.348-4.262/hm 2 , thus the reliability of the model was verified.

3.Lines 50-55: Lack of source references for data listed in the in

---

## [Decision Letter · Decision Letter 1]

31 Oct 2024

PONE-D-24-32522R1Research and Validation of Forest Carbon Sequestration Measurement Model Based on Biomass Method-A Case Study of Guizhou ProvincePLOS ONE

Dear Dr. Yang,

Thank you for submitting your manuscript to PLOS ONE. After careful consideration, we feel that it has merit but does not fully meet PLOS ONE’s publication criteria as it currently stands. Therefore, we invite you to submit a revised version of the manuscript that addresses the points raised during the review process.

The manuscript has been improved, but there are still some technique issues that need to be further addressed.

We look forward to receiving your revised manuscript.

Kind regards,

Dafeng Hui, Ph.D.

Academic Editor

PLOS ONE

Journal Requirements:

Reviewers' comments:

Reviewer's Responses to Questions

**Comments to the Author**

1. If the authors have adequately addressed your comments raised in a previous round of review and you feel that this manuscript is now acceptable for publication, you may indicate that here to bypass the “Comments to the Author” section, enter your conflict of interest statement in the “Confidential to Editor” section, and submit your "Accept" recommendation.

Reviewer #1: (No Response)

Reviewer #2: (No Response)

2. Is the manuscript technically sound, and do the data support the conclusions?

Reviewer #1: (No Response)

Reviewer #2: (No Response)

3. Has the statistical analysis been performed appropriately and rigorously? 

Reviewer #1: (No Response)

Reviewer #2: (No Response)

4. Have the authors made all data underlying the findings in their manuscript fully available?

Reviewer #1: (No Response)

Reviewer #2: (No Response)

5. Is the manuscript presented in an intelligible fashion and written in standard English?

Reviewer #1: (No Response)

Reviewer #2: (No Response)

6. Review Comments to the Author

Reviewer #1: Title of Manuscript: Research and Validation of Forest Carbon Sequestration Measurement Model Based on Biomass Method-A Case Study of Guizhou Province

Overall, I am not very satisfied with the author's point-to-point response. I hope the author can carefully sort out the following issues, otherwise I can only reject the manuscript.

#1. Regarding the fourth review comment, the author did not understand my meaning. What I mean is that the author needs to organize in the introduction rather than just in the discussion.

“4. Lines 57-58: In the introduction, it is mentioned that some traditional research methods have the problems of heavy workload and too much reliance on manual labor, which research methods are specifically referred to, and please give examples of traditional methods in the existing literature to provide an in-depth compendium and summarize the progress.”

#2. The author seems to have omitted a response to the seventh review comment: The specific review comments are as follows:

“7.Lines 435-437 state that this paper derives a formula for sample plot sampling, which, as far as I know, is derived from the methodology for afforestation carbon sink projects (CCER-14-001-V01), what is the difference between the two? Please clarify.”

Reviewer #2: this paper studies the construction of carbon sequestration measurement model based on biomass estimation model. The paper has been well revised, and it is suggested to add some suitable references.

7. PLOS authors have the option to publish the peer review history of their article (what does this mean? ). If published, this will include your full peer review and any attached files.

**Do you want your identity to be public for this peer review?** For information about this choice, including consent withdrawal, please see our Privacy Policy .

Reviewer #1: No

Reviewer #2: No

---

## [Author Response · Author response to Decision Letter 2]

1 Nov 2024

For reviewer one:

1.Regarding the fourth review comment, the author did not understand my meaning. What I mean is that the author needs to organize in the introduction rather than just in the discussion.

“4. Lines 57-58: In the introduction, it is mentioned that some traditional research methods have the problems of heavy workload and too much reliance on manual labor, which research methods are specifically referred to, and please give examples of traditional methods in the existing literature to provide an in-depth compendium and summarize the progress.”

I am sorry that the author did not understand the suggestions of the reviewer at the beginning, and we have supplemented the revision. The traditional method is time-consuming and labor-intensive, and the representative methods are the reserve change method and the profit and loss method. The reserve change method requires that the two survey data are obtained by using the same method, using the same biomass model and parameters, and the data based on continuous sample plots are the most accurate and reliable, but the workload is large, and the accuracy can only be guaranteed on the scale of the sampling population. The profit and loss method obtains the forest carbon sink through the empirical statistical model and the process model, but it has strong regional characteristics, so it needs to re-determine the empirical parameter values or the localized process model parameters when it is applied in other regions, and the parameters of the process model are usually not easy to obtain, which limits the use of the model.

REVISION:

At present, the measurement problems encountered in the development of forestry carbon sequestration are still in the research stage. Some traditional research methods have the problems of heavy workload and over-reliance on manual work, such as the representative reserve change method and profit and loss method [6]. The change in reserves method evaluates the annual change in carbon stocks over time [7]. This method obtains basic data, mainly forest volume, through a large number of ground surveys (such as the National Forest Inventory)[8]. Then the carbon storage of each carbon pool of the ecosystem is estimated by using the established biomass model and carbon measurement parameters (forestry industry standards or related literature), or related equations, biomass carbon density and soil organic carbon density, etc. Finally, the annual change of carbon storage in a period of time is analyzed. If the annual variation of carbon storage is positive, it is regarded as a "carbon sink", otherwise, it is regarded as a "carbon source". Up to 2023, China has carried out several continuous forest resource inventories at intervals of 5 years, and many provinces and regions have also carried out the second class forest resource inventory for 10 years. These survey data and statistical data are the basis of this method.The profit and loss method is to assess the carbon sink function of the ecosystem by studying the carbon budget of the ecosystem in a certain period of time and using the balance results of carbon absorption and emission. This requires a lot of manpower to measure the biomass in the field and establish a regression model with DBH, tree height, volume, forest age and other variables to describe the growth rate. This method can also be used to assess and predict the impact of climate and other environmental factors on growth and carbon accumulation. Commonly used models include FVS [9,10], CACTOS [11], ORGANON [12] and TreeGross [13]. Full carbon pool models mainly include CO2FIX [14], TreeGroSS-C [15], FORCARB [16] and CBM-CFS3 [17-18].The reserve change method requires that the two survey data are obtained by using the same method, using the same biomass model and parameters, and the data based on continuous sample plots are the most accurate and reliable, but the workload is large, and the accuracy can only be guaranteed on the scale of the sampling population. The profit and loss method obtains the forest carbon sink through the empirical statistical model and the process model, but it has strong regional characteristics, so it needs to re-determine the empirical parameter values or the localized process model parameters when it is applied in other regions, and the parameters of the process model are usually not easy to obtain, which limits the use of the model. On the other hand, there is no clear specification for the measurement methods of forest carbon sinks at present. The existing measurement methods mainly include biomass method, volume method and eddy correlation method. Among these methods, the biomass method has high practicability because of its simple and convenient operation, so this paper mainly studies the application of biomass method in the theoretical study of forestry carbon sequestration.

We have also added relevant literature. If the reviewer has any good literature recommendations, please let us know and we will continue to learn

#2. The author seems to have omitted a response to the seventh review comment: The specific review comments are as follows:

“7.Lines 435-437 state that this paper derives a formula for sample plot sampling, which, as far as I know, is derived from the methodology for afforestation carbon sink projects (CCER-14-001-V01), what is the difference between the two? Please clarify.

There is no difference. We use the formula in (CCER-14-001-V01) for reference. We are very sorry that there is no reference. We have made corrections.(reference 21) Thank you again for the professional advice of the reviewer, which is very helpful to us.

For reviewer two:

Thank you very much for your review and suggestions. Wishing you a happy life and smooth work,We have made corresponding modifications to your suggestions in the text

---

## [Editor Report · Decision Letter 2]

11 Nov 2024

Research and Validation of Forest Carbon Sequestration Measurement Model Based on Biomass Method-A Case Study of Guizhou Province

PONE-D-24-32522R2

Dear Dr. Yang,

We’re pleased to inform you that your manuscript has been judged scientifically suitable for publication and will be formally accepted for publication once it meets all outstanding technical requirements.

Kind regards,

Dafeng Hui, Ph.D.

Academic Editor

PLOS ONE
---

## [Editor Report · Acceptance letter]

PONE-D-24-32522R2

PLOS ONE

Dear Dr. Yang,

I'm pleased to inform you that your manuscript has been deemed suitable for publication in PLOS ONE. Congratulations! Your manuscript is now being handed over to our production team.

Kind regards,

on behalf of

Dr. Dafeng Hui

Academic Editor

PLOS ONE